# Expectation Propagation with Stochastic Kinetic Model in Complex Interaction Systems

**Le Fang, Fan Yang, Wen Dong, Tong Guan, and Chunming Qiao**
Department of Computer Science and Engineering
University at Buffalo
{lefang, fyang24, wendong, tongguan, qiao}@buffalo.edu

## Abstract

Technological breakthroughs allow us to collect data with increasing spatio-temporal resolution from complex interaction systems. The combination of high-resolution observations, expressive dynamic models, and efficient machine learning algorithms can lead to crucial insights into complex interaction dynamics and the functions of these systems. In this paper, we formulate the dynamics of a complex interacting network as a stochastic process driven by a sequence of events, and develop expectation propagation algorithms to make inferences from noisy observations. To avoid getting stuck at a local optimum, we formulate the problem of minimizing Bethe free energy as a constrained primal problem and take advantage of the concavity of dual problem in the feasible domain of dual variables guaranteed by duality theorem. Our expectation propagation algorithms demonstrate better performance in inferring the interaction dynamics in complex transportation networks than competing models such as particle filter, extended Kalman filter, and deep neural networks.

## 1 Introduction

We live in a complex world, where many collective systems are difficult to interpret. In this paper, we are interested in complex interaction systems, also called complex interaction networks, which are large systems of simple units linked by a network of interactions. Many research topics exemplify complex interaction systems in specific domains, such as neural activities in our brain, the movement of people in an urban system, epidemic and opinion dynamics in social networks, and so on. Modeling and inference for dynamics on these systems has attracted considerable interest since it potentially provides valuable new insights, for example about functional areas of the brain and relevant diagnoses[7], about traffic congestion and more efficient use of roads [19], and about where, when and to what extent people are infected in an epidemic crisis [23]. Agent-based modeling and simulation [22] is a classical way to address complex systems with interacting components to explore general collective rules and principles, especially in the field of systems biology. However, the actual underlying dynamics of a specific real system are not in the scope. People are not satisfied with only a macroscopic general description but aims to track down an evolving system.

Unprecedented opportunities for researchers in these fields have recently emerged due to the prosperous of social media and sensor tools. For instance, the functional magnetic resonance imaging (fMRI) and the electroencephalogram (EEG) can directly measure brain activity, something never possible before. Similarly, signal sensing technologies can now easily track people's movement and interactions [12, 24]. Researchers no longer need to worry about acquiring abundant observation data, and instead are pursuing more powerful theoretical tools to grasp the opportunities afforded by that data. We, in the machine learning community, are interested in the inference problem — that is

recovering the hidden dynamics of a system given certain observations. However, challenges still exist in these efforts, especially when facing systems with a large number of components.

Statistical inference on complex interaction systems has a close relationship with the statistical physics of disordered ensembles, for instance, the established equivalence between loopy belief propagation and the Bethe free energy formulation [25]. In the past, the main interaction between statistical physics and statistical inference has focused on building stationary and equilibrium probability distributions over the state of a system. However, temporal dynamics is omitted when only equilibrium state is pursued. This leads not only to the loss of a significant amount of interesting information, but possibly also to qualitatively wrong conclusions. In terms of learning dynamics, one approach is to solve stochastic differential equations (SDE) [20]. In each SDE, at least one term belongs to a stochastic process, of which the most common is the Wiener process. The drift and diffusion terms in these SDEs are what we need to recover from multiple realizations (sample paths) of the stochastic process. Typically, an assumption of constant diffusion and linear drift makes the problem tractable, but realistic dynamics generally cannot be modeled by rigid SDEs with simple assumptions.

Inference on complex interaction systems naturally corresponds to inference on large graphical models, which is a classical topic in machine learning. Exact filtering and smoothing algorithms are impractical due to the exploding computational cost to make inferences about complex systems. The hidden Markov model [17] faces an exponentially exploding size of the state transition kernel. The Kalman filter [15] and its variants, such as the extended Kalman filter [14], solves the linear or nonlinear estimation problem assuming that the latent and observed variables are jointly Gaussian distributions. Its scalability versus the number of components is $O(M^3)$ due to the time cost in matrix operations.

Approximate algorithms to make inferences with complex interaction systems can be divided roughly into sampling-based and optimization-based methods. Among sampling based methods, particle filter and smoother [4, 18] use particles to represent the posterior distribution of a stochastic process given noisy observations. However, particle based methods show weak scalability in a complex system: a large number of particles is needed, even in moderate size complex systems where the number of components becomes over thousands. A variety of Markov Chain Monte Carlo (MCMC) methods have been proposed [6, 5], but these generally have issues with rapid convergence in high-dimension systems. Among optimization based methods, expectation propagation (EP) [16, 13] refers to a family of approximate inference algorithms with local marginal projection. These methods adopt an iterative approach to approximate each factor of the target distribution into a tractable family. EP methods have been shown to be relatively efficient, faster than sampling in many low-dimension examples[16, 13]. The equivalence between the EP energy minimization and Bethe free energy minimization is justified [16]. Researches propose "double loop" algorithm to minimize Bethe free energy [13] in order to digest the non-convex term in the objective. They formulate a saddle point problem where strictly speaking the inner loop should be converged before moving to the outer loop. However, the stability of saddle points is an issue in general. There are also ad hoc energy optimization methods for specific network structures, for instance [21] for binary networks, but the generality of these methods is unknown.

In this paper, we present new formulation of EP and apply it to solve the inference problem in general large complex interaction systems. This paper makes the following contributions. First, we formulated expectation propagation as an optimization problem to maximize a concave dual function, where its local maximum is also its global maximum and provides a solution for Bethe free energy minimization problem. To this end, we transformed concave terms in the Bethe free energy into its Legendre dual and added regularization constraint to the primal problem. Second, we designed gradient ascent and fixed point algorithms to make inferences about complex interaction systems with the stochastic kinetic model. In all the algorithms we make mean-field inferences about the individual components from observations about them according to the average interactions of all other components. Third, we conducted experiments on our transportation network data to demonstrate the performance of our proposed algorithms over the state of the art algorithms in inferring complex network dynamics from noisy observations.

The remainder of this paper is organized as follows. In Section 2, we briefly review some models to specify complex system dynamics and the issues in minimizing Bethe free energy. In Section 3, we formulate the problem of minimizing Bethe free energy as maximizing a concave dual function satisfying dual feasible constraint, and develop gradient-based and fixed-point methods to make

tractable inferences with the stochastic kinetic model. In Section 4, we detail empirical results from applying the proposed algorithms to make inferences about transportation network dynamics. Section 5 concludes.

## 2 Background

In this section, we provide brief background about describing complex system dynamics and typical issues in minimizing Bethe free energy.

### 2.1 Dynamic Bayesian Network and State-Space Model

A dynamic Bayesian network (DBN) captures the dynamics of a complex interaction system by specifying how the values of state variables at the current time are probabilistically dependent on the values at previous time. Let $x_t = (x_1^{(1)},..., x_t^{(M)})$ be the values and $y_t = (y_t^{(1)}, y_t^{(2)}, ..., y_t^{(M)})$ be the observations made at these $M$ state variables at time $t$. The probability measure of sample path with observations $p(x_{1,...T}, y_{1,...T})$ can be written as $p(x_{1,...T}, y_{1,...T}) = \prod_t p(x_t \mid x_{t-1})p(y_t \mid x_t) = \prod_t p(x_t \mid x_{t-1}) \prod_m p(y_t^{(m)} \mid x_t^{(m)})$, where $p(x_t \mid x_{t-1})$ is the state transition model and $p(y_t \mid x_t)$ is observation model. We can factorize state transition into miniature kernels involving only variable $x_t^{(m)}$ and its parents $\mathrm{Pa}(x_t^{(m)})$. The DBN inference problem is to infer $p(x_t \mid y_{1,...T})$ for given observations $y_{1,...T}$.

State-space models (SSM) use state variables to describe a system by a set of first-order differential or difference equations. For example, the state evolves as $x_t = F_t x_{t-1} + w_t$ and we make observations with $y_t = H_t x_t + v_t$. Typical filtering and smoothing algorithms estimate series of $x_t$ from time series of $y_t$.

Both DBM and SSM face difficulties in directly capturing the complex interactions, since these interactions seldom obey simple rigid equations and are too complex to be expressed by a joint transition kernel, even allowing time-variance of such kernel. The SKM model that follows uses a sequence of events to capture such nonlinear and time-variant dynamics.

### 2.2 Stochastic Kinetic Model

The stochastic kinetic model (SKM) [9, 23] has been successfully applied in many fields, especially chemistry and system biology [1, 22, 8]. It describes the dynamics with chemical reactions occurring stochastically at an adaptive rate. By analogy with a chemical reaction system, we consider a complex interaction system involving $M$ system components (species) and $V$ types of events (reactions). Generally, the system forms a Markov jump process [9] with a finite set of discrete events. Each event $v$ can be characterized by a "chemical equation":

$$r_v^{(1)} X^{(1)} + ... + r_v^{(M)} X^{(M)} \rightarrow p_v^{(1)} X^{(1)} + ... + p_v^{(M)} X^{(M)} \tag{1}$$

where $X^{(m)}$ denotes the $m$-th component, $r_v^{(m)}$ and $p_v^{(m)}$ count the (relative) quantities of reactants and products. Let $x_t^{(m)}$ be the population count (or continuous number as concentration) of $m$ species at time $t$, an event will change populations $(x_t^{(1)}, x_t^{(2)}, ..., x_t^{(M)})$ by $\Delta_v = (p_v^{(1)} - r_v^{(1)}, p_v^{(2)} - r_v^{(2)}, ..., p_v^{(M)} - r_v^{(M)})$. Events occur mutually independently of each other and each event rate $h_v(x_t, c_v)$ is a function of the current state:

$$h_v(x_t, c_v) = c_v \prod_{m=1}^{(M)} g^{(m)}(x_t^{(m)}) = c_v \prod_{m=1}^{(M)} \binom{x_t^{(m)}}{r_v^{(m)}} \tag{2}$$

where $c_v$ denotes the rate constant and $\prod_{m=1}^{(M)} \binom{x_t^{(m)}}{r_v^{(m)}}$ counts the number of different ways for the components to meet and trigger an event. When we consider time steps $1, 2, ., t, ..T$ with sufficiently small time interval $\tau$, the probability of two or more events happening in the interval is negligible [11]. Consider a sample path $p(x_{1,...T}, v_{2,...T}, y_{1,...T})$ of the system with the sequence of states

$x_1, \ldots, x_T$, happened events $v_2, \ldots, v_T$ and observations $y_1, \ldots, y_T$. We can express the event-based state transition kernel $P(x_t, v_t | x_{t-1})$ in terms of event rate $h_v(x_t, c_v)$:

$$P(x_t, v_t | x_{t-1}) = I\left(x_t = x_{t-1} + \Delta_{v_t} \text{ and } x_t \in (x_{\min}, x_{\max})\right) \cdot P(v_t | x_{t-1})$$

$$= I\left(x_t = x_{t-1} + \Delta_{v_t} \text{ and } x_t \in (x_{\min}, x_{\max})\right) \cdot \begin{cases} \tau h_v(x_{t-1}, c_v) & \text{if } v_t = v \\ 1 - \sum_v \tau h_v(x_{t-1}, c_v) & \text{if } v_t = \emptyset \end{cases} \quad (3)$$

where $\emptyset$ represents a null event that none of those $V$ events happens and states don't change; $I(\cdot)$ is the indicator function; $x_{\min}$, $x_{\max}$ are respectively lower bound and upper bound vectors, which prohibit "ghost" transitions between out-of-scope $x_{t-1}$ and $x_t$. For instance, we generally need to bound $x_t$ be non-negative in realistic complex systems. This natural constraint on $x_t$ leads to a linearly truncated state space that realistic events lie.

Instead of state transitions possibly from any state to any other in DBN and state updates with a linear (or nonlinear) transformation, state in the SKM evolves according to finite number of events between time steps. The transition kernel is dependent on underlying system state and so is adaptive for capturing the underlying system dynamics. We can now consider the inference problem of complex interaction systems in the context of general DBN, with a specific event-based transition kernel from SKM.

## 2.3 Bethe Free Energy

In general DBN, the expectation propagation algorithm to make inference aims to minimize Bethe free energy $F_{\text{Bethe}}$ [16, 25, 13], subject to moment matching constraints. We have a non-convex prime objective and its trivial dual function with dual variables in the full space is not concave. We take the general notation that potential function is $\psi(x_{t-1,t}) = P(x_t, y_t \mid x_{t-1})$ and our optimization problem becomes the following

$$\text{minimize } F_{\text{Bethe}} = \sum_t \int dx_{t-1,t} \hat{p}_t(x_{t-1,t}) \log \frac{\hat{p}_t(x_{t-1,t})}{\psi(x_{t-1,t})} - \sum_t \int dx_t q_t(x_t) \log q_t(x_t)$$

$$\text{subject to}: \langle f(x_t) \rangle_{\hat{p}_t(x_{t-1,t})} = \langle f(x_t) \rangle_{q_t(x_t)} = \langle f(x_t) \rangle_{\hat{p}_{t+1}(x_{t,t+1})}$$

$$\text{maximize } F_{\text{Dual}} = -\sum_t \log \int dx_{t-1,t} \exp(\alpha_{t-1}^\top f(x_{t-1})) \psi(x_{t-1,t}) \exp(\beta_t^\top f(x_t)) + \log \int dx_t \exp((\alpha_t + \beta_t)^\top f(x_t))$$

In the above, $\hat{p}_t(x_{t-1,t}) \approx p(x_{t-1,t} | y_{1,\cdots,T})$ are approximate two-slice probabilities, $q_t(x_t) \approx p(x_t | y_{1,\cdots,T})$ are approximate one-slice probabilities. The vector-valued function $f(x_t)$ maps a random variable $x_t$ to its statistics. Integrals $\langle f(x_t) \rangle_{\hat{p}_t(x_{t-1,t})} = \int dx_t f(x_t) \int dx_{t-1} \hat{p}_t(x_{t-1,t})$ and so on are the mean parameters to be matched in the optimization. $F_{\text{Bethe}}$ is the relative entropy (or K-L divergence) between the approximate distribution $\prod_t \frac{\hat{p}_t(x_{t-1,t})}{q_t(x_t)}$ and the true distribution $p(x_{1,\cdots,T} | y_{1,\cdots,T}) = \prod_t \psi(x_{t-1,t})$ to be minimized. With the method of Lagrange multipliers, one can find that $\hat{p}_t(x_{t-1,t})$ and $q_t(x_t)$ are distributions in the exponential family parameterized either by the mean parameters $\langle f(x_t) \rangle_{\hat{p}_t(x_{t-1,t})}$ and $\langle f(x_t) \rangle_{q_t(x_t)}$ or by the natural parameters $\alpha_{t-1}$ and $\beta_t$, and the trivial dual target $F_{\text{Dual}}$ is the negative log partition of the dynamic Bayesian network.

The problem with minimizing $F_{\text{Bethe}}$ or maximizing $F_{\text{Dual}}$ is that both have multiple local optima and there is no guarantee how closely a local optimal solution approximates the true posterior probability of the latent state. In $F_{\text{Bethe}}$, $\int dx_{t-1,t} \hat{p}_t(x_{t-1,t}) \log \frac{\hat{p}_t(x_{t-1,t})}{\psi(x_{t-1,t})}$ is a convex term, $-\sum_t \int dx_t q_t(x_t) \log q_t(x_t)$ is concave, and the sum is not guaranteed to be convex. Similarly in $F_{\text{Dual}}$, the minus log partition function of $\hat{p}_t$ (first term) is concave, the log partition function of $q_t$ is convex, and the sum is not guaranteed to be concave.

Another difficulty with expectation propagation is that the approximate probability distribution often needs to satisfy some inequality constraints. For example, when approximating a target probability distribution with the product of normal distributions in Gaussian expectation propagation, we require that all factor normal distributions have positive variance. So far, the common heuristic is to set the variances to very large numbers once they fall below zero.

## 3 Methodology

As noted in Subsection 2.3, the difficulty in minimizing Bethe free energy is that both the $F_{\text{Primal}}$ and $F_{\text{Dual}}$ have many local optima in the full space. Our formulation starts with transforming the concave term to its Legendre dual and taking dual variables as additional variables. Thereafter we drop the dependence over $q_t(x_t)$ by utilizing the moment matching constraints, formulate EP as a constrained minimization problem and derive its dual optimization problem (which is concave under a dual feasible constraint). Our formulation also provides theoretical insights to avoid negative variance in Gaussian expectation propagation.

We start by minimizing the Bethe free energy over the two-slice probabilities $\hat{p}_t$ and the one-slice probabilities $q_t$:

minimize over $\hat{p}_t(x_{t-1,t}), q_t(x_t)$ :

$$F_{\text{Bethe}} = \sum_t \int dx_{t-1,t}\hat{p}_t(x_{t-1,t}) \log \frac{\hat{p}_t(x_{t-1,t})}{\psi(x_{t-1,t})} - \sum_t \int dx_t q_t(x_t) \log q_t(x_t)$$

subject to : $\langle f(x_t)\rangle_{\hat{p}_t(x_{t-1,t})} = \langle f(x_t)\rangle_{q_t(x_t)} = \langle f(x_t)\rangle_{\hat{p}_{t+1}(x_{t,t+1})}$ ,

$$\int dx_t q_t(x) = 1 = \int dx_{t-1,t}\hat{p}_t(x_{t-1,t}). \tag{4}$$

We introduce the Legendre dual $-\int dx_t q_t \log q_t = \min_{\gamma_t} \left\{ -\gamma_t^\top \cdot \langle f(x_t)\rangle_{q_t} + \log \int dx_t \exp(\gamma_t^\top \cdot f(x_t)) \right\}$ and replace $\langle f(x_t)\rangle_{q(x_t)}$ in the target with $\langle f(x_t)\rangle_{\hat{p}_t(x_{t-1,t})}$ by utilizing the constraint $\langle f(x_t)\rangle_{\hat{p}_t(x_{t-1,t})} = \langle f(x_t)\rangle_{q_t(x_t)}$. Instead of searching $\gamma_t$ over the over-complete full space, we add a regularization constraint to bound it:

minimize over $\hat{p}_t(x_{t-1,t}), \gamma_t$ :

$$F_{\text{Primal}} = \sum_t \int dx_{t-1,t}\hat{p}_t \log \frac{\hat{p}_t(x_{t-1,t})}{\psi(x_{t-1,t})} - \sum_t \gamma_t^\top \cdot \langle f(x_t)\rangle_{\hat{p}_t} + \sum_t \log \int dx_t \exp(\gamma_t^\top \cdot f(x_t))$$

subject to : $\langle f(x_t)\rangle_{\hat{p}_t(x_{t-1,t})} = \langle f(x_t)\rangle_{\hat{p}_{t+1}(x_{t,t+1})}, \int dx_{t-1,t}\hat{p}_t(x_{t-1,t}) = 1, \gamma_t^\top \gamma_t \leq \eta_t. \tag{5}$$

In the primal problem, $\gamma_t$ is the natural parameter of a probability in the exponential family: $q(x; \gamma_t) = \exp(\gamma_t^\top f(x_t))/\int dx_t \exp(\gamma_t^\top \cdot f(x_t))$. The primal problem (5) is equivalent with Bethe energy minimization problem.

We solve the primal problem with the Lagrange duality theorem [3]. First, we define the Lagrangian function $\mathcal{L}$ by introducing the Lagrange multipliers $\alpha_t$, $\lambda_t$ and $\xi_t$ to incorporate the constraints. Second, we set the derivative over prime variables to zero. Third, we plug the optimum point back into the Lagrangian. The Lagrange duality theorem implies that $F_{\text{Dual}}(\alpha_t, \lambda_t, \xi_t) = \inf_{\hat{p}_t(x_{t-1,t}), \gamma_t} \mathcal{L}(\hat{p}_t(x_{t-1,t}), \gamma_t, \alpha_t, \lambda_t, \xi_t)$. Thus the dual problem is as follows

maximize over $\alpha_t, \lambda_t \geq 0$ for all $t$ :

$$F_{\text{Dual}} = -\sum_t \log Z_{t-1,t} + \sum_t \log \int dx_t \exp(\gamma_t^\top f(x_t)) + \sum_t \frac{\lambda_t}{2} \left(\gamma_t^\top \gamma_t - \eta_t\right) \tag{6}$$

where $-\langle f(x_t)\rangle_{\hat{p}_t} + \langle f(x_t)\rangle_{\gamma_t} + \lambda_t \gamma_t = 0$ \hfill (7)

$$\hat{p}_t(x_{t-1,t}) = \frac{1}{Z_{t-1,t}} \exp(\alpha_{t-1}^\top \cdot f(x_{t-1}))\psi(x_{t-1,t}) \exp((\gamma_t^\top - \alpha_t^\top) \cdot f(x_t)) \tag{8}$$

In the dual problem, we drop the dual variable $\xi_t$ since it takes value to normalize $\hat{p}_t(x_{t-1,t})$ as a valid primal probability. For any dual variable $\alpha_t, \lambda_t$, we map primal variables $\hat{p}_t(x_{t-1,t})$ and $\gamma_t$ as implicit functions defined by the extreme point conditions Eq. (7),(8). We have the following theoretic guarantee with proofs in the supplementary material. We name $\text{cov}_{\gamma_t}(f(x_t), f(x_t)) + \lambda_t I - \langle f(x_t) \cdot f(x_t)^\top\rangle_{\hat{p}_t(x_{t-1,t})} \succ 0$ as the dual feasible constraint.

**Proposition 1: The Lagrangian function has positive definite Hessian matrix under the dual feasible constraint.**

Proposition 1 ensures that the dual function is infimum of Lagrangian function, the point wise infimum of a family of affine functions of $\alpha_t, \lambda_t, \xi_t$, thus is concave. Instead of a full space of dual variables $\alpha_t, \lambda_t$, we only consider the domain constrained by the dual feasible constraint.

**Proposition 2: Eq. (7) and (8) have an unique solution under the dual feasible constraint.**

The Lagrange dual problem is a maximization problem with a bounded domain, which can be reduced to an unconstrained problem through barrier method or through penalizing constraint violation, and be solved with a gradient ascent algorithm or a fixed point algorithm. The partial derivatives of the dual function over dual variables are the following:

$$\frac{\partial F_{\text{Dual}}}{\partial \alpha_t} = -\langle f(x_t)\rangle_{\hat{p}_{t+1}(x_{t,t+1})} + \langle f(x_t)\rangle_{\hat{p}_t(x_{t-1,t})}, \frac{\partial F_{\text{Dual}}}{\partial \lambda_t} = \frac{1}{2}\left(\gamma_t^\top \gamma_t - \eta_t\right) \quad (9)$$

where $\hat{p}_t(x_{t-1,t})$ and $\gamma_t$ are implicit functions defined by Eq. (7),(8). We can get a fixed point iteration through setting the first derivatives to zero [1]. Here $\gamma(\cdot)$ converts mean parameters to natural parameters.

$$\frac{\partial F_{\text{Dual}}}{\partial \alpha_t} \overset{\text{set}}{=} 0 \Rightarrow \text{forward:} \alpha_t^{(\text{new})} = \alpha_t^{(\text{old})} + \gamma\left(\langle f(x_t)\rangle_{\hat{p}_t}\right) - \gamma_t^{(\text{old})}$$

$$\text{backward:} \gamma_t^{(\text{new})} = \gamma\left(\langle f(x_t)\rangle_{\hat{p}_{t+1}}\right)$$

In terms of Gaussian EP, the prime variables $\hat{p}_t(x_{t-1,t})$, $\gamma_t$ correspond to multivariate Gaussian distributions, which pose implicit constraints on the primal and dual domains. Let $\sum_{\hat{p}_t}, \sum_{\gamma_t}$ be the covariance matrix associated with $\hat{p}_t(x_{t-1,t})$, $\gamma_t$ and it requires $\sum_{\hat{p}_t} \succ 0, \sum_{\gamma_t} \succ 0$. The domain of dual variables is defined by the following constraints:

$$\lambda_t \geq 0, \sum_{\hat{p}_t} \succ 0, \sum_{\gamma_t} \succ 0, \text{cov}_{\gamma_t}\left(f(x_t), f(x_t)\right) + \lambda_t I - \langle f(x_t)\cdot f(x_t)^\top\rangle_{\hat{p}_t(x_{t-1,t})} \succ 0$$

$$\text{where } -\langle f(x_t)\rangle_{\hat{p}_t} + \langle f(x_t)\rangle_{\gamma_t} + \lambda_t \gamma_t = 0$$

$$\hat{p}_t(x_{t-1,t}) = \frac{1}{Z_{t-1,t}}\exp(\alpha_{t-1}^\top \cdot f(x_{t-1}))P(x_t, y_t|x_{t-1})\exp((\gamma_t^\top - \alpha_t^\top)\cdot f(x_t))$$

In this case, it is nontrivial to find a starting point of $\alpha_t, \lambda_t$. We develop a phase I stage to find a strictly feasible starting point [3]. For convenience, we note $\alpha_t, \lambda_t$ as $x$, rewrite above constraints as inequality constraints $g_i(x) \leq 0$ and equality constraints $g_j(x) = 0$. Start from a valid $x_0$, $s$ that $g_i(x_0) \leq s, g_j(x_0) = 0$ and then solve the optimization problem

$$\text{minimize } s \text{ subject to } g_j(x_0) = 0, g_i(x_0) \leq s$$

over the variable $s$ and $x$. The strict feasible point of $x$ will be found when we arrive $s < 0$.

With the duality framework and SKM, we can solve the dual optimization problem to make inferences about complex system dynamics from imperfect observations. The latent states (the populations in SKM) can be formulated as either categorical or Gaussian random variables. In categorical case, the statistics are $f(x_t) = (I(x_t^{(1)} = 1), \cdots, I(x_t^{(1)} = x_{\max}^{(1)}), I(x_t^{(2)} = 1), \cdots, I(x_t^{(2)} = x_{\max}^{(2)}), \cdots)$, where $x_{\max}^{(1)}, \cdots, x_{\max}^{(M)}$ are the maximum populations and $I$ is the indicator function. In the Gaussian case, the statistics are $f(x_t) = (x_t^{(1)}, x_t^{(1)\ 2}, x_t^{(2)}, x_t^{(2)\ 2}, \cdots)$ and we force the natural parameters to satisfy the constraint that minus half of precision is negative. The potential $\psi(x_{t-1,t})$ in the

distribution $\hat{p}_{t+1}(x_{t,t+1})$ (Eq. (8)) has specific form $\sum_{v_t} P(x_t, v_t | x_{t-1}) P(y_t | x_t)$ as Eq. (3), which facilitates a mean filed approximation to evaluate $\langle f(x_t) \rangle_{\hat{p}_{t+1}^{(m)}(x_{t,t+1}^{(m)})} \approx \langle f(x_t) \rangle_{\hat{p}_{t+1}(x_{t,t+1})}$ and

$\langle f(x_t) \rangle_{\hat{p}_t^{(m)}(x_{t-1,t}^{(m)})} \approx \langle f(x_t) \rangle_{\hat{p}_t(x_{t-1,t})}$ for each species $m$, where $\hat{p}_{t+1}^{(m)}(x_{t,t+1}^{(m)})$ and $\hat{p}_t^{(m)}(x_{t-1,t}^{(m)})$ are the marginal two-slice distributions for $m$ and derived explicitly in the supplementary material. As such, we establish linear complexity over number of species $m$ and tractable inference in general complex system dynamics.

To summarize, Algorithm 1 gives the mean-field forward-backward algorithm and the gradient ascent algorithm for making inferences with a stochastic kinetic model from noisy observations that minimize Bethe free energy.

---

**Algorithm 1** Make inference of a stochastic kinetic model with expectation propagation.

---

**Input**: Discrete time SKM model (Eqs. (1),(2),(3)); Observation probabilities $P(y_t | x_t)$ and initial values of $\alpha_t, \gamma_t, \lambda_t$ for all populations $m$ and time $t$.
**Expectation Propagation fixed point**: Alternate between forward and backward iterations until convergence.

- For $t = 1, \cdots, T$, $\alpha_t^{(\text{new})} = \alpha_t^{(\text{old})} + \gamma \left( \langle f(x_t) \rangle_{\hat{p}_t(x_{t-1,t})} \right) - \gamma_t^{(\text{old})}$.

- For $t = T, \cdots, 1$, $\gamma_t^{(\text{new})} = \gamma \left( \langle f(x_t) \rangle_{\hat{p}_{t+1}(x_{t,t+1})} \right)$.

**Gradient ascent**: Execute the following updates in alternating forward and backward sweeps, where the gradients are defined in Eq. (9), under the dual feasible constraints.

- $\alpha_t^{(\text{new})} \leftarrow \alpha_t + \epsilon \frac{\partial F_{\text{dual}}}{\partial \alpha_t}$, $\lambda_t^{(\text{new})} \leftarrow \lambda_t + \epsilon \frac{\partial F_{\text{Dual}}}{\partial \lambda_t}$.

**Output**: Optimum $\hat{p}_t(x_{t-1,t})$, $\langle f(x_t) \rangle_{\hat{p}_t}$ as Eq. (7), (8) for all populations $m$ and time $t$.

---

## 4 Experiments on Transportation Dynamics

In this section, we evaluate and benchmark the performance of our proposed algorithms (Algorithm 1) against mainstream state-of-the-art approaches. We have the flexibility to specify species, states, and events with different granularities in SKM, at either macroscopic or microscopic level. Consequently, different levels of inference can be made by feeding in corresponding observations and model specifications. For example, to track epidemics in a social network we can define each person as a species and their health state as a hidden state, with infection and recovery as events. Using real-world datasets about epidemic diffusion in a college campus, we efficiently inferred students' health states compared with ground truth from surveys [23]. In this section, we demonstrate population level inference in the context of transportation dynamics[2].

**Transportation Dynamics** A transportation system consists of residents and a network of locations. The macroscopic description is the number of vehicles indexed by location and time, while the microscopic description is the location of each vehicle at each time. Our goal is to infer the macroscopic populations from noisy sensor network observations made at several selected roads. Such inference problems in complex interaction networks are not trivial, for several reasons: the system can be very large and contain large number of components (residents and locations) and therefore many approaches fail due to resource costs; the interaction between components (i.e. the mobility of residents) is by nature uncertain and time variant, and multiple variables (populations at different locations) correlate together.

To model transportation dynamics, we classify people at the same location as one species. Let $l \in L$ index the locations and $x_t^{(l)}$ be the number of vehicles at location $l$ at time $t$, which are the latent states we want to identify. The events $v$ that change system states can be generally expressed as reaction $l_i \rightarrow l_j$, which represents one vehicle moving from location $l_i$ to location $l_j$. It decrease $x_t^{(l_i)}$ by 1, increase $x_t^{(l_j)}$ by 1 and keep other $x_t^{(l)}$ the same. The event rate reads $h_v(x_t, c_v) = c_v \prod_{l=1}^{(L)} g_v^{(l)}(x_t^{(l)}) = c_v x_t^{(l_i)}$, as there are $x_t^{(l_i)}$ different possible vehicles to transit at $l_i$.

**Experiment Setup:** We select a certain proportion, $e.g.$ 20%, of vehicles as probe vehicles to build the observation model, assuming that the probe vehicles are uniformly sampled from the system. Let $x_{ttl}$ be the total number of vehicles in the system, $x_p$ the total number of probe vehicles, $x_t^{(l)}$ the number of vehicles at location $l$, $y_t^{(l)}$ the number of probe vehicles observed at $l$. A rough point estimation of $x_t^{(l)}$ is $x_t^{(l)} = x_{ttl} y_t^{(l)}/x_p$. More strictly, the likelihood of observing $y_t^{(l)}$ probe vehicles among $x_t^{(l)}$ vehicles at $l$ is $p(y_t^{(l)} \mid x_t^{(l)}) = \binom{x_t^{(l)}}{y_t^{(l)}} \cdot \binom{x_{ttl}-x_t^{(l)}}{x_p - y_t^{(l)}} \Big/ \binom{x_{ttl}}{y_t^{(l)}}$. Our hidden state $x_t^{(l)}$ can be represented as either a discrete variable or a univariate gaussian.

**Dataset Description:** We implement and benchmark algorithms on two representative datasets. In the SynthTown dataset, we synthesize a mini road network (Fig. 1(a)). Virtual residents go to work in the morning and back home in the evening. We synthesize their itineraries from MATSIM, a common Multi-agent transportation simulator[2]. The number of residents and locations are respectively 2,000 and 25. In the Berlin dataset, we have a larger real world road network with 1,539 locations derived from Open Street Map and 9,178 people's itineraries synthesized from MATSIM. Both two datasets span a whole day, from midnight to midnight.

**Evaluation Metrics:** To evaluate the accuracy of the model, we need compare the series of inferred populations against the series of ground truths. We choose three appropriate metrics: the "coefficient of determination" ($R^2$), the mean percentage error (MPE) and mean squared error (MSE). In statistics, the $R^2$ tells the goodness of fit of a model and is calculated as $1 - \frac{\sum_i (y_i - f_i)^2}{\sum_i (y_i - \bar{y})^2}$, where $y_i$ are the ground truth values, $\bar{y}$ their mean and $f_i$ the inferred values. Typically, $R^2$ ranges from 0 and 1: the closer it is to 1, the better the inference is. The MPE computes average of percentage errors by which $f_i$ differ from $y_i$ and is calculated as $\frac{100\%}{n} \sum_i \frac{y_i - f_i}{y_i}$. MPE can be either positive or negative and the closer it is to 0, the better. The MSE is calculated as $\frac{1}{n} \sum_i (y_i - f_i)^2$ to measure the average deviation between $y$ and $f$. The lower the MSE, the better the inference. We also consider the runtime as an important metric to research scalability of different approaches.

**Approaches for Benchmark:** We implement three algorithms to instantiate the procedures in Algorithm 1: the fixed point algorithm with discrete latent state (DFP) or gaussian latent state (GFP) and the gradient ascent algorithm with discrete latent state (DG). The pseudo codes are included in the supplementary material. We also implement several other mainstream state-of-the-art approaches. Particle Filter (PF): We implement a sampling importance resampling (SIR) [10] algorithm that recursively approximates the posterior with a weighted set of particles, updates these particles and resamples to cope with degeneracy problem. Performance is dependent on the number of particles with a certain number is needed to achieve a good result. We selected the number of particles empirically by increasing the number until no obvious accuracy improvement could be detected, and ended up with thousands to tens of thousands of particles. Extended Kalman Filter (EKF): We implement the standard EKF procedure with an alternating prediction step and update step. Feedforward Neural Network (FNN): The FNN builds only a non-parametric model between input nodes and output nodes, without "actually" learning the dynamics of the system. We implement a five-layer FNN: one input layer accepting the inference time point and observations in certain previous period (e.g. one hour), three hidden layers and one output layer from which we directly read the inference populations. The FNN and afterwards RNN are both trained by feeding ground truth populations about each road into the network structures. We tune meta-parameters and train the network with 30 days synthesized mobility data from MATSIM until obtaining optimum performance. Recurrent Neural Network (RNN): The RNN is capable of exploiting previous inferred hidden states recursively to improve current estimation. We implement a typical RNN, such that in each RNN cell we take both the current observations and inferred population from a previous cell as input, traverse one hidden layer, and then output the inferred populations. We train the RNN with 30 days of synthesized mobility data from MATSIM until obtaining optimum performance.

**Inference Performance and Scalability:** Figure 1 plots the inferred population at several representative locations in Fig. 1(a). The lines above the shaded areas are the ground truths, and we plot the error (i.e., inferred populations minus ground truth) with different scales. For GFP, the inference within $\mu \pm 3\sigma$ confidence intervals is shown in the colored "belt". We can see that our proposed algorithms generally deviate less from the ground truth than other approaches do.

Table 1: Performance and time scalability of all algorithms

| Dataset | SynthTown | | | | Berlin | | | |
|---------|-----------|------|------|------|--------|------|------|------|
| Metrics | $R^2$ | MPE | MSE | Time | $R^2$ | MPE | MSE | Time |
| DFP | 0.85 | -3% | 181 | 47 sec | 0.66 | 3% | 20 | 29 min |
| GFP | 0.85 | -8% | 161 | 42 sec | 0.62 | 2.5% | 27 | 21 min |
| DG | 0.87 | -5% | 104 | 157 sec | 0.61 | 2.8% | 26 | 56 min |
| PF | 0.50 | -21% | 663 | 15 sec | 0.50 | -6% | 678 | 71min |
| EKF | 0.51 | -19% | 679 | 2 sec | 0.45 | -40% | 1046 | 14 hour |
| FNN | 0.73 | 11% | 526 | 1 h training | 0.31 | -14% | 540 | 11 h training |
| RNN | 0.72 | -14% | 407 | 8 h training | 0.51 | -9% | 800 | 28 h training |

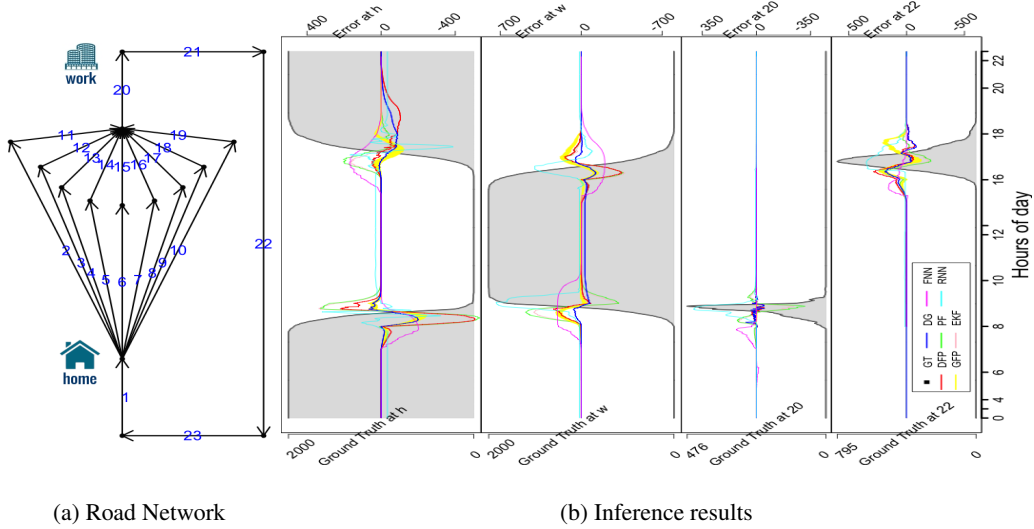

(a) Road Network  (b) Inference results

Figure 1: Road network and inference results with the SynthTown Dataset

Table 1 summarizes the performances in different metrics (mean values). There is both a training phase and a running phase in making inferences with neural networks, with the training phase taking longer. The neural network training time shown in the table ranges from several hours to around one day, and is quadratic in the number of system components per batch per epoch. The neural network running times in our experiments are comparable with EP running times. Theoretically, neural network running times are quadratic in the number of system components to make one prediction, and EP running times are linear in the number of system components to propagate marginal probabilities from one time step to the next (EP algorithms empirically converge within a few iterations), while PF scales quadratically and EKF cubically with the number of locations.

**Summary:** Generally, our proposed algorithms have higher $R^2$, "narrower" MPE and lower MSE, followed by neural networks, PF and EKF. The neural networks sometimes provide comparable performance. Our proposed algorithms, especially the DFP and GFP, experience lower time explosion in bigger datasets. Overall, our algorithms generally outperform PF, EKF, FNN and RNN in terms of accuracy metrics and scalability to a larger dataset.

## 5   Discussion

In this paper, we have introduced the stochastic kinetic model and developed expectation propagation algorithms to make inferences about the dynamics of complex interacting systems from noisy observations. To avoid getting stuck at a local optimum, we formulate the problem of minimizing Bethe free energy as a maximization problem over a concave dual function in the feasible domain of dual variables guaranteed by duality theorem. Our experiments show superior performance over competing models such as particle filter, extended Kalman filter, and deep neural networks.

## Footnotes

[1]Empirically, the fixed point iteration converges even without the dual feasible constraint ($\lambda_t = 0$); In general, $\lambda_t$ is bounded by the dual feasible constraint and the derivative over $\lambda_t$ is not zero.

[2]Source code and a general function interface for other domains at both levels are here online

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
