[Supplementary Material]

## 6 Appendix

### 6.1 Prime optimization problem

In the following, we will derive a constrained optimization problem whose solution minimizes the Bethe free energy (Eq. (13)) under moment matching constraint and additional regularization constraint. The Bethe free energy is convex over $\hat{p}_t$ and concave over $q_t$. Hence it could have multiple minima in the domain of $\hat{p}_t$ and $q_t$. To address this issue, we first introduce the Legendre-Fenchel dual (also called convex conjugate) of $-\int dx_t q_t(x_t) \log q_t(x_t)$ and reformulate the objective function.

We start from minimizing the Bethe free energy $F_{\text{Bethe}}$ subject to the expectation propagation constraints.

minimize over $\hat{p}_t(x_{t-1,t}), q_t(x_t)$ :

$$F_{\text{Bethe}} = \sum_t \int dx_{t-1,t} \hat{p}_t(x_{t-1,t}) \log \frac{\hat{p}_t(x_{t-1,t})}{P(x_t, y_t | x_{t-1})} - \sum_t \int dx_t q_t(x_t) \log q_t(x_t) \qquad (10)$$

subject to :
$$\langle f(x_t) \rangle_{\hat{p}_t(x_{t-1,t})} = \langle f(x_t) \rangle_{q_t(x_t)} = \langle f(x_t) \rangle_{\hat{p}_{t+1}(x_{t,t+1})},$$
$$\int dx_t q_t(x) = 1 = \int dx_{t-1,t} \hat{p}_t(x_{t-1,t}).$$

Formally, the convex conjugate of a function $f(x)$ is defined as $f^*(y) = \max_x \{y^T x - f(x)\}$, where the domain of $y$ is restricted so that the maximum value is finite. This is also known as the Legendre-Fenchel transformation. For each valid distribution $q_t(x_t)$ in the exponential family, the entropy function $-\int dx_t q_t(x_t) \log q_t(x_t)$ can be interpreted as the conjugate function of the log partition:

$$-\int dx_t q_t(x_t) \log q_t(x_t) = \min_{\gamma_t} \left\{ -\gamma_t^\top \cdot \langle f(x_t) \rangle_{q_t} + \log \int dx_t \exp(\gamma_t^\top \cdot f(x_t)) \right\} \qquad (11)$$

The form can be also verified by checking the derivatives over $\gamma_t$. We in essence exploit the Legendre-Fenchel duality between the log partition and the entropy.

We thereafter arrive at

minimize over $\hat{p}_t, q_t, \gamma_t$ for all $t$ :

$$F_{\text{Bethe'}} = \sum_t \int dx_{t-1,t} \hat{p}_t(x_{t-1,t}) \log \frac{\hat{p}_t(x_{t-1,t})}{P(x_t, y_t | x_{t-1})} - \sum_t \gamma_t^\top \cdot \langle f(x_t) \rangle_{q_t} + \sum_t \log \int dx_t \exp(\gamma_t^\top \cdot f(x_t))$$
$$(12)$$

subject to :
$$\langle f(x_t) \rangle_{\hat{p}_t(x_{t-1,t})} = \langle f(x_t) \rangle_{q_t(x_t)} = \langle f(x_t) \rangle_{\hat{p}_{t+1}(x_{t,t+1})},$$
$$\int dx_t q_t(x) = 1 = \int dx_{t-1,t} \hat{p}_t(x_{t-1,t}).$$

To get rid of the dependence over $q(x_t)$, we replace $\langle f(x_t) \rangle_{q(x_t)}$ in the target with $\langle f(x_t) \rangle_{\hat{p}_t(x_{t-1,t})}$ by utilizing the constraint $\langle f(x_t) \rangle_{\hat{p}_t(x_{t-1,t})} = \langle f(x_t) \rangle_{q_t(x_t)}$. Instead of searching $\gamma_t$ over the over-complete whole space, we further add a regularization constraint to bound the prime variable $\gamma_t$ and

will later see how this constraint helps us to build a concave dual function.

minimize over $\hat{p}_t(x_{t-1,t}), \gamma_t$ :

$$F_{\text{Primal}} = \sum_t \int dx_{t-1,t} \hat{p}_t(x_{t-1,t}) \log \frac{\hat{p}_t(x_{t-1,t})}{P(x_t, y_t | x_{t-1})} - \sum_t \gamma_t^\top \cdot \langle f(x_t) \rangle_{\hat{p}_t} + \sum_t \log \int dx_t \exp(\gamma_t^\top \cdot f(x_t))$$

(13)

subject to :

$$\langle f(x_t) \rangle_{\hat{p}_t(x_{t-1,t})} = \langle f(x_t) \rangle_{\hat{p}_{t+1}(x_{t,t+1})}$$

$$\gamma_t^\top \gamma_t \leq \eta_t$$

(14)

$$\int dx_{t-1,t} \hat{p}_t(x_{t-1,t}) = 1.$$

## 6.2 Solving the primal problem with Lagrange duality

We solve this problem with Lagrange duality theorem. First, we define the Lagrangian function $\mathcal{L}$ by introducing the Lagrange multipliers $\alpha_t$, $\lambda_t$ and $\xi_t$ to incorporate those constraints:

$$\mathcal{L} = F_{\text{Primal}} + \sum_t \alpha_t^\top \left( \langle f(x_t) \rangle_{\hat{p}_t(x_{t-1,t})} - \langle f(x_t) \rangle_{\hat{p}_{t+1}(x_{t,t+1})} \right) + \sum_t \frac{\lambda_t}{2} \left( \gamma_t^\top \gamma_t - \eta_t \right)$$

$$+ \sum_t \xi_t \left( \int dx_{t-1,t} \hat{p}_t(x_{t-1,t}) - 1 \right)$$

(15)

where the inequality multiplier $\lambda_t \geq 0$. The Lagrange duality theorem implies that $F_{\text{Dual}}(\alpha_t, \lambda_t, \xi_t) = \inf_{\hat{p}_t(x_{t-1,t}), \gamma_t} \mathcal{L}(\hat{p}_t(x_{t-1,t}), \gamma_t, \alpha_t, \lambda_t, \xi_t)$. To find the infimum of Lagrangian given dual variables, we need first find extreme point of Lagrangian. Set the derivative of $\mathcal{L}$ over $\hat{p}_t(x_{t-1,t}), \gamma_t$ to zero, we get

$$\frac{\partial \mathcal{L}}{\partial \hat{p}_t(x_{t-1,t})} = \log \frac{\hat{p}_t(x_{t-1,t})}{P(x_t, y_t | x_{t-1})} + 1 + \gamma_t^\top \cdot f(x_t) - \alpha_{t-1}^\top \cdot f(x_{t-1}) + \alpha_t^\top \cdot f(x_t) + \xi_t \overset{\text{set}}{=} 0$$

$$\Rightarrow \hat{p}_t(x_{t-1,t}) = \frac{1}{Z_{t-1,t}} \exp(\alpha_{t-1}^\top \cdot f(x_{t-1})) P(x_t, y_t | x_{t-1}) \exp((\gamma_t^\top - \alpha_t^\top) \cdot f(x_t))$$

(16)

where $Z_{t-1,t} = \exp(\xi_t + 1) = \int dx_{t-1,t} \exp(\alpha_{t-1}^\top \cdot f(x_{t-1})) P(x_t, y_t | x_{t-1}) \exp((\gamma_t^\top - \alpha_t^\top) \cdot f(x_t))$

$$\frac{\partial \mathcal{L}}{\partial \gamma_t} = -\langle f(x_t) \rangle_{\hat{p}_t} + \langle f(x_t) \rangle_{\gamma_t} + \lambda_t \gamma_t = 0$$

(17)

where $\langle f(x_t) \rangle_{\gamma_t} = \frac{\int dx_t f(x_t) \exp(\gamma_t^\top f(x_t))}{\int dx_t \exp(\gamma_t^\top f(x_t))}$

Our notation with $\gamma_t$ as subscript means the statistics over the exponential family distribution parameterized by $\gamma_t$. Substituting Eq. (16) into our Lagrangian function (Eq. (15)), we get the following dual form, which is concave over $\alpha_t, \lambda_t$ for all $t$. This is an concave maximization problem whose solution is the global maximum.

maximize over $\alpha_t, \lambda_t$ for all $t$ :

$$F_{\text{Dual}} = -\sum_t \log Z_{t-1,t} + \sum_t \log \int dx_t \exp(\gamma_t^\top f(x_t)) + \sum_t \frac{\lambda_t}{2} \left(\gamma_t^\top \gamma_t - \eta_t\right) \tag{18}$$

subject to : $\lambda_t \geq 0$

where $Z_{t-1,t} = \int dx_{t-1,t} \exp(\alpha_{t-1}^\top \cdot f(x_{t-1})) P(x_t, y_t | x_{t-1}) \exp((\gamma_t^\top - \alpha_t^\top) \cdot f(x_t))$ (19)

$$-\langle f(x_t)\rangle_{\hat{p}_t} + \langle f(x_t)\rangle_{\gamma_t} + \lambda_t \gamma_t = 0 \tag{20}$$

$$\hat{p}_t(x_{t-1,t}) = \frac{1}{Z_{t-1,t}} \exp(\alpha_{t-1}^\top \cdot f(x_{t-1})) P(x_t, y_t | x_{t-1}) \exp((\gamma_t^\top - \alpha_t^\top) \cdot f(x_t)) \tag{21}$$

In the dual problem, we have dropped the dual variable $\xi_t$ since it takes value to normalize $\hat{p}_t(x_{t-1,t})$ as a valid primal probability. For any dual variable $\alpha_t, \lambda_t$, we have mapped primal variables $\hat{p}_t(x_{t-1,t})$ and $\gamma_t$ as implicit functions defined by the extreme point conditions Eq. (16),(17). We have the following theoretic guarantee.

**Proposition 1: The Lagrangian function has positive definite Hessian matrix when**
$\mathbf{cov}_{\gamma_t}\left(f(x_t), f(x_t)\right) + \lambda_t I - \left\langle f(x_t) \cdot f(x_t)^\top \right\rangle_{\hat{p}_t(x_{t-1,t})} \succ 0.$

$\boldsymbol{Proof}$ : The hessian matrix is defined as a square matrix of second-order partial derivatives over variables. Since the variables are all indexed by time $t$ and there is no correlation term between two variables indexed with $t$ and $t'$. It's suffice to check the positive definiteness of Hessian over one time slice, i.e. over $\hat{p}_t(x_{t-1,t}), \gamma_t$ . We can finally claim overall positive definiteness by noticing that a sum of positive semi-definite matrix with non-intersect column vectors $z$ to make $z^T M z = 0$ will be a positive definite matrix.

With the form of Lagrangian in Eq. (15), the hessian matrix over $\hat{p}_t(x_{t-1,t}), \gamma_t$ becomes

$$\mathbf{H} = \left[ \begin{array}{cc} \frac{1}{\hat{p}_t(x_{t-1,t})} & -f(x_t) \\ -f(x_t)^\top & \mathbf{cov}_{\gamma_t}\left(f(x_t), f(x_t)\right) + \lambda_t \end{array} \right]$$

Using Schur complements, we have the equivalence condition of above hessian matrix to be positive definite as:

$$\mathbf{H} \succ 0 \iff \mathbf{cov}_{\gamma_t}\left(f(x_t), f(x_t)\right) + \lambda_t I - \left\langle f(x_t) \cdot f(x_t)^\top \right\rangle_{\hat{p}_t(x_{t-1,t})} \succ 0$$

Thus the proof is done.$\square$

The Proposition 1 ensures the dual function as infimum of Lagrangian function given dual variable. Since the dual function is the point wise infimum of a family of affine functions of $\alpha_t, \lambda_t, \xi_t$, it is concave. We name $\mathbf{cov}_{\gamma_t}\left(f(x_t), f(x_t)\right) + \lambda_t I - \left\langle f(x_t) \cdot f(x_t)^\top \right\rangle_{\hat{p}_t(x_{t-1,t})} \succ 0$ as dual feasible constraint. Instead of a whole space of dual variables $\alpha_t, \lambda_t$, now we only consider constrained domain by dual feasible constraint.

**Proposition 2: The implicit function of $\hat{p}_t(x_{t-1,t})$ and $\gamma_t$ defined by Eq. (16), (17) has unique solution under dual feasible constraint.**

$\boldsymbol{Proof}$ : The extreme point equations define implicit function of $\hat{p}_t(x_{t-1,t})$ and $\gamma_t$. Consider $-\langle f(x_t)\rangle_{\hat{p}_t} + \langle f(x_t)\rangle_{\gamma_t} + \lambda_t \gamma_t = 0$ and plug in Eq. (16), we have $\gamma_t$ as root of function $F(\gamma_t) = -\langle f(x_t)\rangle_{\hat{p}_t} + \langle f(x_t)\rangle_{\gamma_t} + \lambda_t \gamma_t$. Check the derivative, we have $\frac{\partial F(\gamma_t)}{\partial \gamma_t} = -\text{Var}_{\hat{p}_t}(f(x_t)) + \mathbf{cov}_{\gamma_t}\left(f(x_t), f(x_t)\right) + \lambda_t I$. The dual feasible constraint is $\mathbf{cov}_{\gamma_t}\left(f(x_t), f(x_t)\right) + \lambda_t I - \left\langle f(x_t) \cdot f(x_t)^\top \right\rangle_{\hat{p}_t(x_{t-1,t})} \succ 0$. Therefore we have $\mathbf{cov}_{\gamma_t}\left(f(x_t), f(x_t)\right) + \lambda_t I \succ \left\langle f(x_t) \cdot f(x_t)^\top \right\rangle_{\hat{p}_t(x_{t-1,t})} \succ \text{Var}_{\hat{p}_t}(f(x_t))$ and $\frac{\partial F(\gamma_t)}{\partial \gamma_t} \succ 0$.

For monotonic functional $F(\gamma_t)$, it has at most one root. Since $F(\gamma_t)$ could achieve negative/positive infinity when $\gamma_t$ takes negative/positive infinity, we have the root of $F(\gamma_t) = 0$ has unique solution. $\square$

The Lagrange dual problem is a concave maximization problem with bounded domain. Hence it has a unique global optima. A gradient ascent algorithm or a converging fixed point algorithm should converge to the solution. The partial derivatives of the dual function over the dual variables are the following.

$$
\begin{aligned}
\frac{\partial F_{\text{Dual}}}{\partial \alpha_t} &= -\langle f(x_t)\rangle_{\hat{p}_{t+1}(x_{t,t+1})} + \langle f(x_t)\rangle_{\hat{p}_t(x_{t-1,t})} + \frac{\partial \gamma_t}{\partial \alpha_t} \cdot \left( -\langle f(x_t)\rangle_{\hat{p}_t} + \langle f(x_t)\rangle_{\gamma_t} + \lambda_t \gamma_t \right) \\
&= -\langle f(x_t)\rangle_{\hat{p}_{t+1}(x_{t,t+1})} + \langle f(x_t)\rangle_{\hat{p}_t(x_{t-1,t})} \\
\frac{\partial F_{\text{Dual}}}{\partial \lambda_t} &= \frac{1}{2}\left( \gamma_t^\top \gamma_t - \eta_t \right) + \frac{\partial \gamma_t}{\partial \lambda_t} \cdot \left( -\langle f(x_t)\rangle_{\hat{p}_t} + \langle f(x_t)\rangle_{\gamma_t} + \lambda_t \gamma_t \right) \\
&= \frac{1}{2}\left( \gamma_t^\top \gamma_t - \eta_t \right)
\end{aligned}
$$

where $\hat{p}_t(x_{t-1,t})$ and $\gamma_t$ are implicit functions defined by the extreme point conditions Eq. (16),(17). Hence we can get a fixed point iteration through the first derivatives over $\alpha_t$ to zero. Empirically, the fixed point iteration converges even without the dual feasible constraint ($\lambda_t = 0$); Since the dual feasible constraint bound the $\lambda_t$, we should not set the derivative over $\lambda_t$ to zero.

$$
\begin{aligned}
\frac{\partial F_{\text{Dual}}}{\partial \alpha_t} \overset{\text{set}}{=} 0 \Rightarrow \text{forward:} & \alpha_t^{(\text{new})} = \alpha_t^{(\text{old})} + \gamma\left( \langle f(x_t)\rangle_{\hat{p}_t(x_{t-1,t})} \right) - \gamma_t^{(\text{old})} \\
\text{backward:} & \gamma_t^{(\text{new})} = \gamma\left( \langle f(x_t)\rangle_{\hat{p}_{t+1}(x_{t,t+1})} \right)
\end{aligned}
$$

## 6.3 Inference with SKM

In the SKM, we have the event based kernel as Eq. (3) and the form of $\hat{p}_t(x_{t-1,t})$ as Eq. 16. We write $\gamma_t - \alpha_t$ as $\beta_t$ and make mean field assumption that $\alpha_{t-1}^\top \cdot f(x_{t-1}) = \sum_m \alpha_{t-1}^{(m)T} \cdot f(x_{t-1}^{(m)})$, $\beta_t^\top \cdot f(x_t) = \sum_m \beta_t^{(m)T} \cdot f(x_t^{(m)})$, where the parameter $\alpha_{t-1}^{(m)}, \beta_t^{(m)}$, the statistics $f(x_{t-1}^{(m)})$, $f(x_t^{(m)})$ only involve one specific species $m$ and there is no correlation terms. Substitute the $P(x_t, v_t|x_{t-1})$ explicitly and , i.e. , we have

$$
\begin{aligned}
\hat{p}_t(x_{t-1,t}, v_t) = & \frac{1}{Z_{t-1,t}} \prod_m \alpha_{t-1}^{(m)}(x_{t-1}^{(m)}) \prod_m \beta_t^{(m)}(x_t^{(m)}) \cdot P(y_t|x_t) \cdot I(x_t \in (x_{\min}, x_{\max})) \\
& \cdot \begin{cases} \tau \cdot c_v \prod_{m=1}^M g_v^{(m)}(x_{t-1}^{(m)}) \cdot \prod_{m=1}^M I(x_t^{(m)} - x_{t-1}^{(m)} = \Delta_v^{(m)}) & \text{if } v_t = v \\ (1 - \tau \sum_v c_v \prod_{m=1}^M g_v^{(m)}(x_{t-1}^{(m)})) \cdot \prod_{m=1}^M I(x_t^{(m)} - x_{t-1}^{(m)} = 0) & \text{if } v_t = \emptyset \end{cases}
\end{aligned}
\tag{22}
$$

To simplify the notation, we abbreviate $\exp(\alpha_{t-1}^{(m)T} \cdot f(x_{t-1}^{(m)}))$ as $\alpha_{t-1}^{(m)}(x_{t-1}^{(m)})$ and $\exp(\beta_t^{(m)T} \cdot f(x_t^{(m)}))$ as $\beta_t^{(m)}(x_t^{(m)})$. We can marginalize the joint solution Eq. 22 over $x_{t-1}^{(m')}, x_t^{(m')}$ for all the other species $m' \neq m$ and get the marginal distribution for each particular species $m$:

For $v_t = v$,

$$\hat{p}_t^{(m)}(x_{t-1}^{(m)}, x_t^{(m)}, v_t) = \frac{1}{Z_{t-1,t}} \alpha_{t-1}^{(m)}(x_{t-1}^{(m)}) P(y_t^{(m)}|x_t^{(m)}) \beta_t^{(m)}(x_t^{(m)}) \cdot I\left(x_t^{(m)} \in (x_{\min}^{(m)}, x_{\max}^{(m)})\right)$$

$$\cdot \tau c_v g_v^{(m)}(x_{t-1}^{(m)}) I(x_t^{(m)} - x_{t-1}^{(m)} = \Delta_v^{(m)})$$

$$\cdot \prod_{m' \neq m} \int_{x_t^{(m')} - x_{t-1}^{(m')} = \Delta_v^{(m')}} dx_{t-1,t}^{(m')} \alpha_{t-1}^{(m')}(x_{t-1}^{(m')}) P(y_t^{(m')}|x_t^{(m')}) \beta_t^{(m')}(x_t^{(m')}) g_v^{(m')}(x_{t-1}^{(m')}) I\left(x_t^{(m')} \in (x_{\min}^{(m')}, x_{\max}^{(m')})\right)$$

For $v_t = \emptyset$,

$$\hat{p}_t^{(m)}(x_{t-1}^{(m)}, x_t^{(m)}, v_t) = \frac{1}{Z_{t-1,t}} \alpha_{t-1}^{(m)}(x_{t-1}^{(m)}) P(y_t^{(m)}|x_t^{(m)}) \beta_t^{(m)}(x_t^{(m)}) \cdot I\left(x_t^{(m)} \in (x_{\min}^{(m)}, x_{\max}^{(m)})\right)$$

$$\cdot I(x_t^{(m)} - x_{t-1}^{(m)} = 0)$$

$$\cdot \prod_{m' \neq m} \int_{x_t^{(m')} - x_{t-1}^{(m')} = 0} dx_{t-1,t}^{(m')} \alpha_{t-1}^{(m')}(x_{t-1}^{(m')}) P(y_t^{(m')}|x_t^{(m')}) \beta_t^{(m')}(x_t^{(m')}) I\left(x_t^{(m')} \in (x_{\min}^{(m')}, x_{\max}^{(m')})\right)$$

$$- \frac{1}{Z_{t-1,t}} \alpha_{t-1}^{(m)}(x_{t-1}^{(m)}) P(y_t^{(m)}|x_t^{(m)}) \beta_t^{(m)}(x_t^{(m)}) \cdot I\left(x_t^{(m)} \in (x_{\min}^{(m)}, x_{\max}^{(m)})\right)$$

$$\cdot \tau c_v g_v^{(m)}(x_{t-1}^{(m)}) I(x_t^{(m)} - x_{t-1}^{(m)} = 0)$$

$$\cdot \prod_{m' \neq m} \int_{x_t^{(m')} - x_{t-1}^{(m')} = 0} dx_{t-1,t}^{(m')} \alpha_{t-1}^{(m')}(x_{t-1}^{(m')}) P(y_t^{(m')}|x_t^{(m')}) \beta_t^{(m')}(x_t^{(m')}) g_v^{(m')}(x_{t-1}^{(m')}) I\left(x_t^{(m')} \in (x_{\min}^{(m')}, x_{\max}^{(m')})\right)$$

Extract the term $\prod_{m' \neq m} \int_{x_t^{(m')} - x_{t-1}^{(m')} = 0} dx_{t-1,t}^{(m')} \alpha_{t-1}^{(m')}(x_{t-1}^{(m')}) P(y_t^{(m')}|x_t^{(m')}) \beta_t^{(m')}(x_t^{(m')}) I\left(x_t^{(m')} \in (x_{\min}^{(m')}, x_{\max}^{(m')})\right)$, we arrive at

$$\hat{p}_t^{(m)}(x_{t-1}^{(m)}, x_t^{(m)}, v_t) = \frac{1}{Z_t^{(m)}} \cdot \alpha_{t-1}^{(m)}(x_{t-1}^{(m)}) P(y_t^{(m)}|x_t^{(m)}) \beta_t^{(m)}(x_t^{(m)}) \cdot P(x_t^{(m)}, v_t|x_{t-1}^{(m)})$$

$$where \; P(x_t^{(m)}, v_t|x_{t-1}^{(m)}) = I\left(x_t^{(m)} \in (x_{\min}^{(m)}, x_{\max}^{(m)})\right) \cdot$$

$$\begin{cases} c_v \tau g_v^{(m)}(x_{t-1}^{(m)}) \prod_{m' \neq m} \tilde{g}_v^{(m')} \cdot I(x_t^{(m)} - x_{t-1}^{(m)} = \Delta_v^{(m)}) & v_t^{(m)} = v \\ \left(1 - \sum_v c_v \tau g_v^{(m)}(x_{t-1}^{(m)}) \prod_{m' \neq m} \hat{g}_v^{(m')}\right) I(x_t^{(m)} - x_{t-1}^{(m)} = 0) & v_t^{(m)} = \emptyset \end{cases}$$

$$\tilde{g}_v^{(m')} = \frac{\int_{x_t^{(m')} - x_{t-1}^{(m')} = \Delta_v^{(m')}} dx_{t-1,t}^{(m')} \alpha_{t-1}^{(m')}(x_{t-1}^{(m')}) P(y_t^{(m')}|x_t^{(m')}) \beta_t^{(m')}(x_t^{(m')}) g_v^{(m')}(x_{t-1}^{(m')}) I\left(x_t^{(m')} \in (x_{\min}^{(m')}, x_{\max}^{(m')})\right)}{\int_{x_t^{(m')} - x_{t-1}^{(m')} = 0} dx_{t-1,t}^{(m')} \alpha_{t-1}^{(m')}(x_{t-1}^{(m')}) P(y_t^{(m')}|x_t^{(m')}) \beta_t^{(m')}(x_t^{(m')}) I\left(x_t^{(m')} \in (x_{\min}^{(m')}, x_{\max}^{(m')})\right)}$$

$$\hat{g}_v^{(m')} = \frac{\int_{x_t^{(m')} - x_{t-1}^{(m')} = 0} dx_{t-1,t}^{(m')} \alpha_{t-1}^{(m')}(x_{t-1}^{(m')}) P(y_t^{(m')}|x_t^{(m')}) \beta_t^{(m')}(x_t^{(m')}) g_v^{(m')}(x_{t-1}^{(m')}) I\left(x_t^{(m')} \in (x_{\min}^{(m')}, x_{\max}^{(m')})\right)}{\int_{x_t^{(m')} - x_{t-1}^{(m')} = 0} dx_{t-1,t}^{(m')} \alpha_{t-1}^{(m')}(x_{t-1}^{(m')}) P(y_t^{(m')}|x_t^{(m')}) \beta_t^{(m')}(x_t^{(m')}) I\left(x_t^{(m')} \in (x_{\min}^{(m')}, x_{\max}^{(m')})\right)} \tag{23}$$

$$Z_t^{(m)} = \frac{Z_t}{\prod_{m' \neq m} \int_{x_t^{(m')} - x_{t-1}^{(m')} = 0} dx_{t-1,t}^{(m')} \alpha_{t-1}^{(m')}(x_{t-1}^{(m')}) P(y_t^{(m')}|x_t^{(m')}) \beta_t^{(m')}(x_t^{(m')}) I\left(x_t^{(m')} \in (x_{\min}^{(m')}, x_{\max}^{(m')})\right)}$$

$$= \int_{x_t^{(m)} = x_{t-1}^{(m)}} dx_{t-1,t}^{(m)} \alpha_{t-1}^{(m')}(x_{t-1}^{(m')}) P(y_t^{(m')}|x_t^{(m')}) \beta_t^{(m')}(x_t^{(m')}) I\left(x_t^{(m')} \in (x_{\min}^{(m')}, x_{\max}^{(m')})\right)$$

$$\cdot \left(\sum_v c_v \tau \prod_{all \; m} \tilde{g}_v^{(m)} + 1 - \sum_v c_v \tau \prod_{all \; m} \hat{g}_v^{(m)}\right)$$

, where $Z_t^{(m)}$, $Z_t$ are respectively the normalization constant of $\xi_t^{(m)}(x_{t-1}^{(m)}, x_t^{(m)}, v_t)$ and $\xi_t(x_{t-1}, x_t, v_t)$. In Eq. 23, $\hat{p}_t^{(m)}(x_{t-1}^{(m)}, x_t^{(m)}, v_t)$ takes the same form as the joint solution Eq. 22, except a marginalized transition kernel $P(x_t^{(m)}, v_t | x_{t-1}^{(m)})$ which sums over all the other species $m' \neq m$. Instead of coping with exploding joint state space, we can now cope with each marginalized Markov chain with kernel $P(x_t^{(m)}, v_t | x_{t-1}^{(m)})$. Moreover, $\tilde{g}_v^{(m')}$, $\hat{g}_v^{(m')}$ can be interpreted as expectation of $g$ factor at species $m'$. This suggests that each species evolves their states marginally according to the average effects of the others.

To summarize, we give the general algorithms.

---

**Algorithm 2** Fixed Point Algorithm

---

**Input:** The discrete time SKM model (Eq. 1, 2, 3); the observations $y_t^{(m)}$ for all $t$, $m$; the observation model $P(y_t^{(m)} | x_t^{(m)})$; any initialization of $\alpha_t^{(m)}, \beta_t^{(m)}, \lambda_t^{(m)} > 0$

1: Define function: $\underline{\textbf{ForwardTransition}(\alpha_{t-1}^{(m)}, \beta_t^{(m)})}$
   Find $\xi_t^{(m)}(x_{t-1,t}^{(m)})$ from Eq. 23; Find $\gamma_t^{(m)}$ from $\left\langle f(x_t^{(m)}) \right\rangle_{\gamma_t^{(m)}(x_t^{(m)})} = \left\langle f(x_t^{(m)}) \right\rangle_{\xi_t^{(m)}(x_{t-1,t}^{(m)})}$;
   Update $\alpha_t^{(m)} \leftarrow \gamma_t^{(m)} - \beta_t^{(m)}$. $\underline{\textbf{Output } \alpha_t^{(m)}, \gamma_t^{(m)}}$

2: Define function: $\underline{\textbf{BackwardTransition}(\alpha_{t-1}^{(m)}, \beta_t^{(m)})}$
   Find $\xi_t^{(m)}(x_{t-1,t}^{(m)})$ from Eq. 23; Find $\gamma_{t-1}^{(m)}$ from $\left\langle f(x_{t-1}^{(m)}) \right\rangle_{\gamma_{t-1}^{(m)}(x_{t-1}^{(m)})} = \left\langle f(x_{t-1}^{(m)}) \right\rangle_{\xi_t^{(m)}(x_{t-1,t}^{(m)})}$;
   Update $\beta_{t-1}^{(m)} \leftarrow \gamma_{t-1}^{(m)} - \alpha_{t-1}^{(m)}$. $\underline{\textbf{Output } \beta_{t-1}^{(m)}, \gamma_{t-1}^{(m)}}$

3: **repeat**
4:   **for** t=2 to T **do**
5:     $\alpha_t^{(m)}, \gamma_t^{(m)} \leftarrow$ **ForwardTransition**($\alpha_{t-1}^{(m)}, \beta_t^{(m)}$)
6:   **end for**
7:   **for** t=T-1 to 1 **do**
8:     $\beta_t^{(m)}, \gamma_t^{(m)} \leftarrow$ **BackwardTransition**($\alpha_t^{(m)}, \beta_{t+1}^{(m)}$)
9:   **end for**
10: **until** Convergence of $\gamma_t^{(m)}$
11: Output $\gamma_t^{(m)}$

---

---

**Algorithm 3** Gradient Ascent Algorithm

---

**Input:** The discrete time SKM model (Eq. 1, 2, 3); the observations $y_t^{(m)}$ for all $t$, $m$; the observation model $P(y_t^{(m)} | x_t^{(m)})$; any initialization of $\alpha_t^{(m)}, \beta_t^{(m)}, \gamma_t^{(m)} = \alpha_t^{(m)} + \beta_t^{(m)}$; Function **ForwardTransition**, **BackwardTransition** in algorithm 2

1: **repeat**
2:   **for** t=2 to T-1 **do**
3:     **repeat**
4:       Update $\hat{p}_t(x_{t-1,t}) = \frac{1}{Z_{t-1,t}} \exp(\alpha_{t-1}^\top \cdot f(x_{t-1})) P(x_t, y_t | x_{t-1}) \exp((\gamma_t^\top - \alpha_t^\top) \cdot f(x_t))$
         and $\langle f(x_t) \rangle_{\hat{p}_t}$ according to 7,8,9 under dual feasible constraint
5:     **until** Convergence or enough number of iterations
6:   **end for**
7:   **for** t=T-1 to 2 **do**
8:     Do the same as line 4 to 6
9:   **end for**
10: **until** Convergence
11: Output $\hat{p}_t(x_{t-1,t}) = \frac{1}{Z_{t-1,t}} \exp(\alpha_{t-1}^\top \cdot f(x_{t-1})) P(x_t, y_t | x_{t-1}) \exp((\gamma_t^\top - \alpha_t^\top) \cdot f(x_t))$ and $\langle f(x_t) \rangle_{\hat{p}_t}$

---