[Reviews · NeurIPS 2017]

Reviewer 1



This paper considers the variational approach to the inference problem on certain type of temporal graphical models. It defines a convex formulation of the Bethe free energy, resulting in a method with optimal convergence guarantees. The authors derive Expectation-Propagation fixed point equations and gradient-based updates for solving the optimization problem. A toy example is used to illustrate the approach in the context of transportation dynamics and it is shown that the proposed method outperforms sampling, extended Kalman Filtering and a neural network method. In the variational formulation, the optimization problem is generally non-convex, due to the entropy terms. Typical variational methods bound the concave part linearly and employ an EM-like algorithm. The key contribution of this paper is to work out a dual formulation on the natural parameters so that the dual problem becomes convex. The derivations seem correct and this looks like an important contribution, if I have not missed some important detail. Authors should elaborate on the generality and applicability of their approach to other models. I think it would be useful to compare their method with the standard double-loop approach [11] both in terms of describing the differences and in numerical comparison (in terms of error and cpu-time). The authors state that they are minimizing the Bethe free energy, in found a bit confusing that they call this a mean field forward and backward in the algorithm. Regarding the experiments, the state representation goes from a factorized state with the state of an agent being one component to location states in which the identities of the agents are lost and the components become strongly coupled. This is OK, but to me it contradicts a bit the motivation in the introduction, in which the authors state "we are not satisfied with only a macroscopic description of the system (...) the goal is (..) to track down the actual underlying dynamics". Some details about the other methods would be good to know. For example, how many particles N where used for PF? How sensitive are the results (accuracy and time) w.r.t N? Also it is not clear how the NNs are trained. Is the output provided the actual hidden state? The paper looks too dense. I think substantial part in the Introduction is not necessary and more space could be gain so that formulas do not need to be in reduced font and figure 1 can be made larger. Other minor comments: It should be stated in the beginning that x is continuous variable unless otherwise stated Eq (3) no need to define g_v, since it is not used in the rest of the paper line 130: better $h(x_{t-1},c_v)$ than $h(x_{t},c_v)$ Eq (4) I think $1-\tau$ should be $(1-\tau)$ Equations between 139-140. \psi(|x_{t-1,1}) -> \psi(x_{t-1,1}) Equations between 139-140. "F_dual" -> "F_Dual" line 104: "dynamics" -> "dynamic" line 126: remove "we have that" line 193: -> "by strong duality both solutions exist and are unique" line 194: "KKT conditions gives" -> "KKT conditions give" Use (???) instead of ??? for equations Algorithm 1: the name can be improved Algorithm 1: Gradient ascent: missing eqrefs in Eqs. ... Algorithm 1: Gradient ascent: missing parenthesis in all $\langle f(x_t \rangle$ Figure 1 line 301: "outperforms against" -> "outperform" line 302: "scales" -> "scale"

Reviewer 2



Overall a well-written and interesting paper in the domain of nonlinear state-space models. The nonlinear state-space dynamics follow a particular prior determined by the "stochastic kinetic model". Exact Inference in the model is intractable due to nonlinearities so Expectation Propagation is used to infer p(x|y) and the predict future observations. Excellent results are obtained for the problem at hand. I'm particularly impressed with how much better EP-based algorithms do w.r.t. RNNs for this problem. The analysis for runtime is weak however, the scaling analysis is missing. It would be nice to have an explanation as to why PF and EKF scale so badly and show it in a separate plot. Also, why are runtime values missing for RNNs? This is a big issue. You cannot claim your method scales better than RNNs and then not give runtime values for RNNs ;) Another overall issue I have with this paper is that the inductive bias implicit in the choice of prior for the dynamics may cause this method to not work well at all with time series from other domains. A few more experiments showing performance on datasets from other domains is essential for this paper to be considered as an accept. Overall I do like the approach and love that EP is beating neural nets. I just want some more extensive experiments.

Reviewer 3



This paper claims to make a new setup for doing EP that guarantees the local optima of the energy function is also a global optima. The introduction reads as if they have solve this for the general case. However, it appears this is only done for the stochastic kinetic models case. In any case, this is a big claim. The derivations to back up this claim on page 5 are dense. I did not have time check them entirely. The introduction reads seems long and a bit disorganized. A shorter introduction with a more clear train of thought would be better. Clarity would also be increased by stating the probability of everything for the SKM as was done for the state space model in Equation 1. Questions Equation 1 seems to be basically the definition of a state space model. Wouldn't it be simpler to state it as such? Typos: L28, L137 Latex issues: L127, L161